# Development of a Tandem Mass Spectral Library for the Detection of Triterpenoids in Plant Metabolome Based on Reference Standards

**DOI:** 10.3390/plants13233278

**Published:** 2024-11-22

**Authors:** Bibi Zareena, Syed Usama Y. Jeelani, Adeeba Khadim, Arslan Ali, Jalal Uddin, Satyajit D. Sarker, Matthias Rainer, Shaden A. M. Khalifa, Hesham R. El-Seedi, Muhammad Ramzan, Syed Ghulam Musharraf

**Affiliations:** 1H.E.J. Research Institute of Chemistry, International Center for Chemical and Biological Sciences, University of Karachi, Karachi 75270, Pakistan; zarina_ku@yahoo.com (B.Z.); syedusamayaseen@gmail.com (S.U.Y.J.); adeeba.abbas@gmail.com (A.K.); rafridi73@gmail.com (M.R.); 2Dr. Panjwani Center for Molecular Medicine and Drug Research, International Center for Chemical and Biological Sciences, University of Karachi, Karachi 75270, Pakistan; arslanali1986@gmail.com; 3Department of Pharmaceutical Chemistry, College of Pharmacy, King Khalid University, Abha 61421, Saudi Arabia; jalaluddinamin@gmail.com; 4School of Pharmacy and Biomolecular Sciences, Liverpool John Moores University, James Parsons Building Byrom Street, Liverpool L3 3AF, UK; s.sarker@ljmu.ac.uk; 5Institute for Analytical Chemistry and Radiochemistry, University of Innsbruck, A-6020 Innsbruck, Austria; m.rainer@uibk.ac.at; 6Neurology and Psychiatry Department, Capio Saint Göran’s Hospital, Sankt Göransplan 1, 112 19 Stockholm, Sweden; 7International Research Center for Food Nutrition and Safety, Jiangsu University, Zhenjiang 212013, China; 8Department of Chemistry, Faculty of Science, Islamic University of Madinah, Madinah 42351, Saudi Arabia

**Keywords:** triterpenoids, electrospray ionization–mass spectrometry, generation of spectral library, ESI-QTOF-MS/MS method

## Abstract

Plant triterpenoids represent a diverse group of secondary metabolites and are thought to be valuable for therapeutic applications. For drug development, lead optimization, better knowledge of biological pathways, and high-throughput detection of secondary metabolites in plant extracts are crucial. This paper describes a qualitative method for the rapid and accurate identification of various triterpenoids in plant extracts using the LC-HR-ESI-MS/MS tool in combination with the data-dependent acquisition (DD) approach. A total of 44 isolated, purified, and characterized triterpenoids were analyzed. HR-MS spectra and tandem mass spectra (MS/MS) of each compound were recorded in the positive ionization mode in two different sets of collisional energies, i.e., (25–62.5 eV), and fixed collisional energies (10, 20, 30, and 40 eV). As a result, three triterpenoids were identified in all plant extracts using the retention time, high-resolution mass spectra, and/or MS/MS spectra. The developed method will be helpful with other plant extracts/botanicals, as well as in the search for new triterpenoids in the kingdom Plantae.

## 1. Introduction

Plants have the ability to produce a variety of specialized metabolites. Among them, triterpenoids are a significant group of plant secondary metabolites that play roles not only in the defense and development of plants but also have potential applications in the food and pharmaceutical industries [1]. Based on the diverse features in their structures, triterpenoids are categorized into different subgroups: acyclic, monocyclic, bicyclic, tricyclic, tetracyclic, and pentacyclic [2]. Among them, pentacyclic triterpenoids (PCTTs) have gained significant attention due to their effects in promoting health and being a type of natural active substance [3,4]. At present, numerous pentacyclic triterpenes have been identified and classified based on the carbon skeleton into four subgroups, namely oleanane, ursane, lupane, and friedelane types [3,5].

Pentacyclic triterpenoid’s medicinal values have been increasingly recognized in recent years [4], thus being introduced as a pharmaceutical for the treatment of various ailments [4,6,7,8,9,10,11,12,13,14,15,16,17,18,19]. Considering the pharmacological properties of triterpenoids, screening plant extracts and products to identify and characterize them is extremely important. Nowadays, the most used technology for phytochemical analysis is liquid chromatography–high-resolution mass spectrometry (LC-HR-MS).

Compounds can be identified using reference standards and/or spectral comparison. Numerous efforts have been made to create MS/MS libraries for different classes and subclasses of natural products. For example, the saponins mass spectrometry database (SMSD) was created using 214 reference compounds from commercial sources and is useful for the rapid identification of saponins [20]. Similarly, a database for quinoline alkaloids was established using data from electrospray ionization–tandem mass spectrometry (ESI-MSn) [21]. Recently, we have curated a spectral library of alkaloids that includes metadata, such as precursor and fragment ions, along with the compound name and their retention times [22]. The WEIZMASS spectral library [23], National Institute of Standard and Technology (NIST) [24], Global Natural Products Social molecular networking (GNPS) [25], Mass Bank of North America (MoNA) [26], etc., are some examples of commercial and public sources of MS/MS spectral databases. However, based on chromatographic and mass spectrum characteristics, there is currently no such LC-ESI-MS/MS database available that can be used to quickly and accurately identify biologically significant triterpenoids.

In this study, a set of 44 standard plant triterpenoids that were purified and structurally verified were used to develop an LC-ESI-MS/MS method employing the data-dependent acquisition (DDA) approach. In DDA, ions are continuously selected and separated from the full scan spectrum (MS1) to generate MS/MS spectra [27]. DDA is widely used to detect known metabolites in mixtures like plant extracts and plasma [28]. A spectral database of triterpenoids, along with information about their precursor ions, mass fragments, retention times, exact masses, and other metadata, is provided using the established approach. The method validity and practical applicability were evaluated, using retention times of reference triterpenoids, the MS/MS mode of analysis at average collision energy (CE), and targeted MS/MS analysis by searching for triterpenoids in diverse plant extracts. This study might play a significant role in the high-throughput and cost-effective identification of triterpenoids in several plant extracts and complex herbal formulations.

## 2. Results and Discussion

### 2.1. Liquid Chromatography–Mass Spectrometry Analysis

To separate plant triterpenoids, a reversed-phase (RP-C18) column with a linear gradient was used in the liquid chromatographic procedure. All the detected reference standard compounds were eluted within the analysis time. The compounds were grouped based on Log *p* values because this value suggests hydrophobicity and aids in preventing compound co-elution during LC separation. A study by Nir Shahaf et al. used a pooling strategy with 20 compounds [23]. Additionally, the majority of metabolite profiling studies used columns with diameters of 2.1 mm × 100 mm and peak capacities of >200 [29]. Therefore, we suggested employing more compounds in a pool with several chromatographic parameters to boost the high-throughput analysis of the triterpenoids in complex samples. This number (44 in the current analysis of triterpenoids) was higher than in the earlier investigations [23,30] but similar to our previously published study [22]. The resulting triterpenoids are given in Table 1, including compounds name, exact masses, molecular formula, and Log-*p* values along with detected parameters; however, the structures are illustrated in Appendix A. The implementation of this pooling strategy leads to a significant reduction in analysis time and an improvement in cost-effectiveness when compared to previously conducted studies that depended on the injection of a single sample.

### 2.2. Optimization of MS/MS Spectral Features

The MS/MS patterns created in the QTOF collisional cell through the fragmentation of compounds depend on the energy applied in the collision cell and are indicative of structural properties. To acquire tandem mass spectral data, two different MS/MS data acquisition techniques were used, namely the auto MS/MS mode using an increase in energy from 25 to 62.5 eV and the targeted MS/MS mode using a precursor list of compounds at fixed collision energies (10, 20, 30, and 40 eV). Individual analytical runs were undertaken for the two acquisitions, and the generated data of the spectral database for triterpenoids are compiled in Table 1. However, [M + H]^+^ ions dominated in the positive ionization mode; other ions in the mass spectrum revealed the presence of sodium adducts. The resulting detected ionic species *m*/*z* are also mentioned in Table 1. Electrospray ionization is a softer ionization process than other mass spectrometry methods, as it produces fewer fragments [31], and in this line, our library data are presented in Figure 1.

**Figure 1 plants-13-03278-f001:**
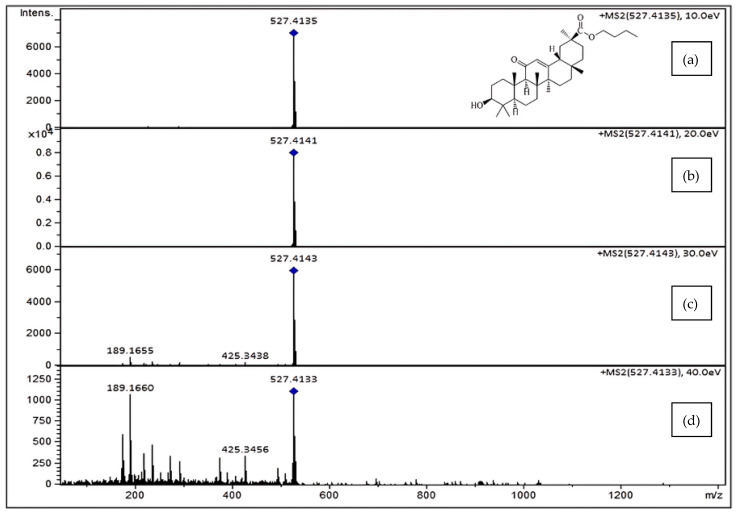
MS2 spectra of butyl ester of glycyrrhetinic acid at various collision energies: (**a**) 10 eV, (**b**) 20 eV, (**c**) 30 eV, and (**d**) 40 eV.

**Table 1 plants-13-03278-t001:** Triterpenoids identified in various plant extracts using the developed triterpenoid library.

S. No.	Compound Name	Log-*p* Values	Molecular Formula	RT (min)	Adduct Identified	*m*/*z* calc.	*m*/*z* meas.	Error (ppm)
1	5-Hydroxy-7-{{6-O-{[(4R/S)-4-(1-hydroxy-1-methylethyl)cyclohex-1-en-1-yl]-carbonyl}c-D-glucopyranosyl}oxy}-2-methyl-4H-1-benzopyran-4-one	2.83	C_26_H_32_O_11_	7.39	[M + H]^+^	521.2017	521.2045	−5.3
					[M + Na]^+^	543.1837	543.1867	−5.5
2	11-Oxooleanolic acid	6.78	C_30_H_46_O_4_	9.76	[M + H]^+^	471.3469	471.349	−4.5
3	Asiatic acid	6.46	C_30_H_48_O_5_	9.8	[M + H]^+^	489.3575	489.356	2.9
					[M + Na]^+^	511.3394	511.339	0.8
4	Euscaphic acid	6.21	C_30_H_48_O_5_	9.11	[M + H]^+^	489.3575	489.356	2.9
					[M + Na]^+^	511.3394	511.338	2.6
5	Silymin A	4.92	C_30_H_44_O_5_	9.9	[M + H]^+^	485.3262	485.3249	2.6
					[M + Na]^+^	507.3081	507.3038	8.4
6	β-Neriursate	11.47	C_38_H_54_O_4_	10.35	[M + H]^+^	575.4095	575.4124	−5.1
7	3β-Hydroxy-27-p-E-coumaroyloxy-urs-12-en-28-oic acid	9.47	C_39_H_54_O_6_	10.02	[M + H]^+^	619.3993	619.3984	1.5
					[M + Na]^+^	641.3813	641.3800	2.0
8	Ilelatifol D	6.29	C_30_H_46_O_4_	9.11	[M + H]^+^	471.3469	471.3455	2.9
9	Glycyrrhetic acid	6.57	C_30_H_46_O_4_	9.86	[M + H]^+^	471.3469	471.3465	0.8
10	Bellerigenin B	3.65	C_30_H_48_O_7_	8.84	[M + H]^+^	521.3473	521.3466	1.3
					[M + Na]^+^	543.3292	543.3287	1
11	Intybusoloid	2.24	C_27_H_34_O_8_	8.83	[M + H]^+^	487.2326	487.2322	0.9
					[M + Na]^+^	509.2146	509.2141	1
12	Ursolaldehyde	9.14	C_30_H_48_O_2_	11.43	[M + H]^+^	441.3727	441.3711	3.7
					[M + Na]^+^	463.3547	463.3536	2.3
13	Hydrazide of glycyrrhetinic acid	5.28	C_32_H_50_N_2_O_5_	9.63	[M + H]^+^	543.3792	543.3769	4.4
					[M + Na]^+^	565.3612	565.3586	4.5
14	Ethyl methyl sulfide ester of betulinic acid	10.27	C_33_H_54_O_3_S_1_	10.36	[M + H]^+^	531.3866	531.3863	0.6
15	2, 3, 23-Triacetoxy derivative of asiatic acid	8.59	C_36_H_54_O_8_	10.19	[M + H]^+^	615.3891	615.3878	2.2
					[M + Na]^+^	637.3711	637.3695	2.5
16	Betulinic acid	8.94	C_30_H_48_O_3_	10.58	[M + H]^+^	457.3676	457.3664	2.6
					[M + Na]^+^	479.3496	479.3473	4.7
17	Betulin	9.01	C_30_H_50_O_2_	10.35	[M + H]^+^	443.3884	443.3858	5.7
					[M + Na]^+^	465.3703	465.3690	2.8
18	Lantanilic acid	8.37	C_35_H_52_O_6_	10.43	[M + H]^+^	569.3837	569.3827	1.7
					[M + Na]^+^	591.3656	591.3669	−2.1
19	3-Benzoyloxy-3-0-methyl ester of glycyrrhetinic acid	9.91	C_38_H_52_O_5_	10.89	[M + H]^+^	589.3888	589.3866	3.7
					[M + Na]^+^	611.3707	611.3697	1.6
20	Methyl acetate ester of betulinic acid	9.50	C_33_H_52_O_5_	10.87	[M + H]^+^	529.3888	529.3902	−2.6
					[M + Na]^+^	551.3707	551.3721	−2.5
21	Azadiradione	4.21	C_28_H_34_O_5_	8.54	[M + H]^+^	451.2479	451.2486	−1.5
					[M + Na]^+^	473.2298	473.2306	−1.6
22	Gedunin	3.34	C_28_H_34_O_7_	8.82	[M + H]^+^	483.2377	483.2388	−2.2
					[M + Na]^+^	505.2197	505.2206	−1.9
23	Ethyl acetate ester of glycyrrhetinic acid	7.63	C_34_H_52_O_6_	10.33	[M + H]^+^	557.3837	557.3845	−1.6
					[M + Na]^+^	579.3656	579.3665	−1.5
24	Oleanolic acid	9.06	C_30_H_48_O_3_	10.51	[M + H]^+^	457.3676	457.3654	4.8
					[M + Na]^+^	479.3496	479.3504	−1.7
25	3-Oxo-30-butyl ester of glycyrrhetinic acid	8.35	C_34_H_52_O_4_	10.94	[M + H]^+^	525.3938	525.3952	−2.7
					[M + Na]^+^	547.3758	547.3772	−2.5
26	β-amyrin	11.06	C_30_H_50_O	10.50	[M + H]^+^	427.3934	427.3920	3.3
					[M + Na]^+^	449.3754	449.3730	5.3
27	Diethyl sulfide ester of betulinic acid	10.80	C_34_H_56_O_3_S_1_	10.69	[M + H]^+^	545.4023	545.403	−1.3
28	Nimbinolide	0.71	C_30_H_36_O_11_	7.51	[M + H]^+^	573.233	573.232	1.8
					[M + Na]^+^	595.215	595.2139	−1.8
29	Ursonic acid	8.43	C_30_H_46_O_3_	10.81	[M + H]^+^	455.3520	455.3512	1.7
					[M + Na]^+^	477.3339	477.3333	1.4
30	3-Acetoxy-3-0-methyl ester of glycyrrhetinic acid	7.89	C_33_H_50_O_5_	10.88	[M + H]^+^	527.3731	527.3722	1.7
					[M + Na]^+^	549.355	549.3542	1.5
31	3,25-Epoxy-3α-hydroxy-olean-12-en-28-oic acid	7.29	C_30_H_46_O_4_	10.62	[M + H]^+^	471.3469	471.3451	3.8
					[M + Na]^+^	493.3288	493.3286	0.5
32	Butyl ester of glycyrrhetinic acid	8.59	C_34_H_54_O_4_	10.89	[M + H]^+^	527.4095	527.4097	−0.4
					[M + Na]^+^	549.3914	549.3917	−0.5
33	Methyl acetate ester of glycyrrhetinic acid	7.10	C_33_H_50_O_6_	10.2	[M + H]^+^	543.368	543.3683	−0.6
					[M + Na]^+^	565.35	565.3502	−0.5
34	Glycyrrhetinic acid methyl ester	7.00	C_31_H_48_O_4_	10.32	[M + H]^+^	485.3625	485.3615	2.1
					[M + Na]^+^	507.3445	507.3446	−0.3
35	Diethyl sulfide ester of ursolic acid	10.87	C_34_H_56_O_3_S_1_	10.58	[M + H]^+^	545.4023	545.4021	0.3
36	Oleanoic acid	8.48	C_30_H_46_O_3_	10.79	[M + H]^+^	457.3676	457.3635	8.9
					[M + Na]^+^	477.3339	477.334	−0.2
37	Methyl acetate ester of oleanolic acid	9.62	C_33_H_52_O_5_	10.93	[M + Na]^+^	551.3707	551.3708	−0.2
38	3-Oxolup-1:12-diene, 28-al	8.05	C_30_H_44_O_2_	10.56	[M + Na]^+^	459.3234	459.3279	−10
39	3β-Hydroxyurs-11-en-13b(28)-olide	7.48	C_30_H_46_O_3_	10.53	[M + H]^+^	455.352	455.3539	0.6
					[M + Na]^+^	477.3339	477.3315	5.0
40	2|A-Hydroxyursolic acid	7.82	C_30_H_48_O_4_	10.1	[M + H]^+^	473.3625	473.3615	2.1
					[M + Na]^+^	495.3445	495.3486	−8.2
41	Friedelin	10.87	C_30_H_50_O	10.60	[M + H]^+^	427.3934	427.3926	1.8
					[M + Na]^+^	449.3754	449.3740	3.1
42	Ursolic acid	9.01	C_30_H_48_O_3_	10.7	[M + H]^+^	457.3676	457.3659	3.7
					[M + Na]^+^	479.3496	479.3459	7.7
43	Atriplicin	7.06	C_30_H_46_O_4_	9.8	[M + H]^+^	471.3469	471.3451	3.8

Using Bruker Compass LibraryEditor (4.4), both [M + H]^+^ and [M + Na]^+^ mass spectral data, as well as their additional metadata for all the analyzed standards, were assembled into a spectral database. A reference standard triterpenoid entry in the developed spectral library is shown in Figure 2D. This spectral library will be especially helpful for comparing compounds with good peak intensities of their sodium adducts during the identification process.

### 2.3. MS/MS Spectral Features of Standard Triterpenoids

Among 44 triterpenoids under study, 18 were oleanane, 14 were ursanes, 6 were lupanes, and 1 was a friedelane; there were also 4 tetranortriterpenoid compounds of limonoid type and 1 belonging to an unknown group. These compounds exhibit distinct fragmentation behavior, which helps to provide more information for fragment matching in the library. Triterpenoid compounds of the oleanane type, such as glycyrrhetic acid, which has a hydroxyl substituent at carbon number 3 (C-3), prominently lose a water molecule [M + H − H_2_O]^+^. All oleanane-type compounds with a carboxyl substituent present in their structure displayed an [M + H − HCOOH]^+^ ion and/or peak of [M+ H − H_2_O − HCOOH]^+^. Another oleanane-type triterpenoid, lantanilic acid sodium adduct, produced peaks at *m*/*z* 491.3195 and 545.3714 as a result of C_5_H_8_O_2_ and HCOOH losses, which are comparable to its [M + H]^+^ fragments previously described. However, this sodium adduct does not show any peak for [M + H − H_2_O]^+^ [32].

Pentacyclic triterpenoids majorly exhibit a common fragmentation, i.e., the retro-Diels–Alder (RDA) fragmentation pathway [33]. The peak at *m*/*z* 249.1856 is directly produced from [M + H]^+^ ion RDA fragmentation in the case of simple skeletons with a hydroxy group at carbon number-3 (C-3) and carboxyl substituent at C-22, whereas the fragments at *m*/*z* 191.1803 and 203.1807 appeared from RDA fragmentation, i.e., [M + H − 2H_2_O − CO]^+^ and [M + H − H_2_O]^+^, respectively. Due to RDA fragmentation, compounds with a dimethylacroyloxy substituent present in the skeleton produced just a few or no fragments at all.

Ursane triterpenoids such as ilelatifol D, with two hydroxyl groups at C-2 and C-3, produce fragment [M + H − H_2_O − CO]^+^ at *m*/*z* 425 through the simultaneous loss of water and the CO molecule at a CE of 20 eV. Similarly, all ursane triterpenoids possessing a carboxylic group in their structure produced fragments [M + H − HCOOH]^+^ and/or [M + H − H_2_O − HCOOH]^+^ due to the loss of formic acid, similar to oleanane-type compounds. Peaks resulting from direct RDA fragmentation and loss from precursor ions are both common for ursane-type triterpenoids. In some ursane-type compounds, the base peak was the product ion at *m*/*z* 205, which is the result of the RDA fragmentation of these compounds. These compounds’ MS/MS spectra also contain peaks at *m*/*z* 203. Triterpenoids of the lupane type displayed significant substituent losses in addition to peaks at *m*/*z* 191 and 203. These triterpenoids, due to lack of unsaturation in their skeleton, showed the given fragments due to pathways other than RDA. All these types of pentacyclic triterpenoids showed similar mass spectral features as reported previously [32]. Additionally, it was found that the fragmentation behavior of limonoids, a class of nortriterpenoids, was similar to what had been reported in the literature [34].

### 2.4. Screening of Plant Extracts Against Library

With the aid of established search parameters, the proposed approach is used to rapidly identify triterpenoids in different plant extracts. By using the compound MS(n) tool, the features were generated from five different plants. The chromatograms, RT, and MS/MS spectra of each precursor detected in plant extracts against the created library, as seen in Figure 2A–C, depict the results of the triterpenoid MS/MS library search for compound matching the purposes. Consequently, the identification of the compound β-neriursate, found in an extract of *Peganum harmala,* was reported. A total of three compounds were found in five different plant extracts. All three compounds were validated based on their fragment match, retention time, and exact masses in a mass tolerance window (0.005 Da), with standard compounds (analyzed under identical conditions) already curated in the spectral library, fulfilling the highest identification level (Level 01) in metabolomics described by AC-Schrimpe-Rutledge et al., and Jody C. May et al. [35]. The maximum retention time drift for the detected compounds was 0.41 min. Table 2 lists the specific compounds identified in each plant extract through a search of the developed library.

## 3. Materials and Methods

### 3.1. Chemicals and Reagents

The chemicals and solvents used in this study were either of analytical reagent grade or HPLC grade. Ultrapure deionized water (resistivity 18.1 MΩ cm at 25 °C) was acquired from the Barnstead MicroPure Purification System (Thermo Scientific, Waltham, MA, USA). The Molecular Bank of Dr. Panjwani Center for Molecular Medicine and Drug Research, International Center for Chemical and Biological Sciences, University of Karachi, Pakistan, provided the triterpenoid standards (purity ≥ 98) of plant sources.

### 3.2. Standard Solution Preparation

Stock solutions of standard triterpenoids were primarily made in methanol, while a few solutions were made in acetone or chloroform and stored below 0 °C. These pre-determined plant-based metabolites were combined into a pool consisting of 44 standards of varying lipophilicity. The pool was made by combining an equal volume (10 μL) of each standard’s stock solution, preparing a dilution of 10 times for the pool. Before injecting the standard pool into the LC-MS instrument, it was filtered through a Millipore filter (0.22 µm). ACD Lab software (version 2081.1) was used to calculate the Log *p* values of all standards.

### 3.3. Sample Solution Preparation

Different plant extract (*Peganum harmala* L., *Camellia sinensis* (L.) Kuntze, *Aegle marmelos* L., *Adhatoda vasica* (L.) Nees, and *Papaver somniferum* L.) solutions were made using 500 g of dried, uniformly powdered whole plants that were extracted overnight on a shaker at 70 rpm in 1 L of methanol/water (1:1) followed by filtration. A vacuum drying procedure was used at 45 °C to concentrate the extracts. The stock solution of concentration 1 mg/mL in acetonitrile was made and diluted 10X. After centrifuging the diluted solutions at 12,000 rpm for 20 min, the supernatants were filtered through a Millipore filter (0.22 µm) and transferred to individual autosampler vials for analysis.

### 3.4. LC-MS and MS/MS Analysis

Ultra-high-performance liquid chromatographic and mass spectrometry, i.e., the Bruker maXis II ESI-HR-QTOF (Bruker Daltonics, Bremen, Germany) instrument was used to analyze both triterpenoid standards and samples as described previously [22]. Injection volumes used for standards and extracts were 5 and 2 μL, respectively. Eluent A, water, and B, methanol, both containing 0.1% formic acid, served as the mobile phases. A gradient elution program consisting of a flow rate of 0.5 mL/min was used. The chromatographic process was started using a reverse-phase C18 column, Macherey-Nagel Nucleodur Gravity (2.0 × 100 mm, 1.8 µm), and guard column (2.0 × 4 mm, 1.8 µm), with initializing conditions of 5% B, elevated to 95% B in 9 min, held for 1 min, and then reset to the initial setup for equilibration until 13 min. Triterpenoid standards and samples’ ESI-MS data were acquired at room temperature in the positive ionization mode. The nebulizer and collision gases were both highly pure nitrogen gases. The following values are listed as the parameters for ESI mass spectral data acquisition: capillary voltage at 4500 V, nebulizer gas at 2.8 bar, endplate offset at 500 V, drying gas temperature at 300 °C, drying gas flow rate at 10.0 L/min, MS scan speed set at 5 Hz, and MS/MS analysis at 12 Hz for spectra rate. The scan range for the time-of-flight (TOF) value was set to 50–1200 amu. Triterpenoid standards underwent two types of MS2 analysis: auto MS/MS mode and targeted MS/MS mode using a scheduled precursor list (SPL). The optimal number of active exclusions was set at 3, and an intensity threshold of 1000 counts was set to start the acquisition of the precursor for the formation of fragment ions. In contrast, a fixed set of collision energies was employed in the targeted method (10, 20, 30, and 40 eV).

### 3.5. Data Processing Procedure

Before each run, a calibrant (sodium formate, 10 mM) was injected at a rate of 3 μL/min for mass scale calibration. The Compass TargetAnalysis software version 1.3 (Bruker Daltonik) was used to perform targeted screening of the acquired data. This targeted analyzed data file is subsequently processed to obtain detailed MS2 information for each standard triterpenoid using Compass DataAnalysis 4.4 software (Bruker Daltonik). Following that, all standard triterpenoid MS and MS/MS spectra were combined using Library Editor 4.4 (Bruker) to create a mass spectral database. For plant samples, raw data were calibrated by a calibrant followed by compound–MS(n) generation before being searched against the resulting library.

## 4. Conclusions

This study provided a rapid and qualitative identification of triterpenoids in different plant extracts based on HR-ESI-MS, RT, and MS/MS data. The data-dependent acquisition (DDA) method based on a tandem mass spectral library of 44 chiefly unique triterpenoids standards with significant biological values was curated, containing data on the standards’ monoisotopic mass, retention time, compound name, chemical formula, MS, and MS/MS spectra. A cost-effective and time-saving pooling method was used for the analysis of up to 44 triterpenoids in a pool to efficiently utilize LC-MS runtime. To make compound identification even easier, standardized LC retention times were also added to the library. Five medicinal plant extracts were searched against the standard data to determine the (LC-MS/MS) method applicability. Based on reference standards retention time, HR-MS, and/or MS/MS spectra, three different triterpenoids were identified. This study may be helpful for the rapid identification of already known triterpenoids and for discovering new biologically active triterpenoids in complex plant extracts by using the generated data as a reference.

## Figures and Tables

**Figure 2 plants-13-03278-f002:**
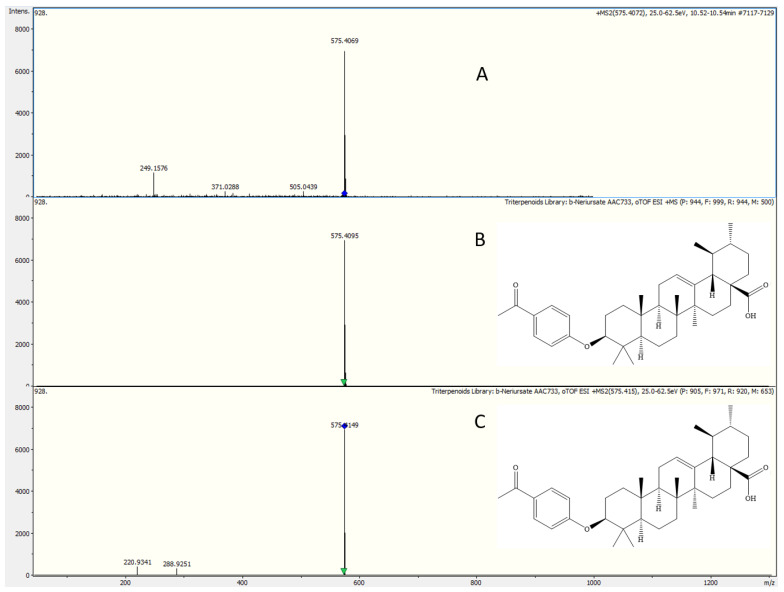
(**A**) β-Neriursate, identified in Peganum Harmala, extracts raw data using the constructed Library. (**B**) Reference standard β-Neriursate MS spectra in the constructed library. (**C**) MS2 spectra of standard β-neriursate in the constructed library. (**D**) Depiction of β-neriursate library record in Bruker Library Editor 4.

**Table 2 plants-13-03278-t002:** List of triterpenoids identified in various plant extracts using the developed triterpenoid library.

Source	Compound Name	RT [min]	Drift RT [min]	Fragments Ions
***Peganum harmala* L.**	β-Neriursate	10.53	0.2	220.9341
***Camellia sinensis* Kuntze**	Butyl ester of glycyrrhetinic acid	11.18	0.3	425.3459, 288.9212, 220.9335, 189.1656, 175.1472, 149.0973
***Aegle marmelos* L.**	β-Neriursate_[Na]_	9.94	0.41	509.2737, 288.9202, 259.1144, 215.0896, 171.0639, 155.0685
***Adhatoda Vasica* L.**	Silymin_[Na]_	9.95	0.08	421.2227, 390.2389, 243.1232, 155.0704
***Papaver somniferum* L.**	Silymin A_[Na]_	9.97	0.06	155.0693

## Data Availability

Data are contained within the article.

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
