# Peer review of "Development of a Tandem Mass Spectral Library for the Detection of Triterpenoids in Plant Metabolome Based on Reference Standards"

_plants, 2024, doi:10.3390/plants13233278_

Round 1
Reviewer 1 Report (Previous Reviewer 2)
Comments and Suggestions for Authors
Here are my comments on manuscript entitled “Development of Tandem Mass Spectral Library for the Detection of Triterpenoids in Plant Metabolome based on Reference Standards”.
The manuscript is submitted under a new ID number. However, the authors have provided their replies based on their previous submission of the same manuscript. I see substantial improvement of the manuscript and to my opinion, it is in acceptable form for publication.
Reviewer 2 Report (Previous Reviewer 1)
Comments and Suggestions for Authors
Authors:
I have read and reviewed your revised manuscript on Development of tandem mass spectral library for the detection of triterpenoids. You have replied to all my previous queries and criticism. The level of the manuscript was improved.
The Introduction is fine, and it gives adequate information on the state-of-the-art in the area, and give a definition of the current investigation and the objectives of the work presented in the manuscript. During the revision process it was slightly changed.
The Results and Discussion were also improved. The formerly criticized Table 1 and Table 2 were re-formated, and the Tables look much better now. The criticized errors were corrected. Figure 2 was also improved.
Experimental part: I understand the answer on HR-MS and MS/MS data, and the absent NMR data what I criticized before. This item was answered and I agree.
In summary: The revised manuscript was substantially improved in all criticized items. Its present version is eligible for being accepted for publication.
This manuscript is a resubmission of an earlier submission. The following is a list of the peer review reports and author responses from that submission.
Round 1
Reviewer 1 Report
Comments and Suggestions for Authors
Authors:
I have read and reviewed your manuscript on Development of tandemmass spectral library for the detection of triterpenoids.
The Introduction is fine, and it gives adequate information on the state-of-the-art in the area, and give a definition of the current investigation and the objectives of the work presented in the manuscript.
The Results and Discusion are correct in principle. However, most of the figures and/or tables presented therein are of a poore quality.
Table 1: Only the chemical names of compounds No. 1 and 2 are long, the rest of the chemical names are quite short. Therefore, I see no reason, why the column called Compound name should be so wide? If this column is made narrower, then the columns in the right side of the table could be made wider (all of them), and numbers presented there would appear in one line. At present, the data in this table are very difficult to read and understand. The Table should be re-modelled and improved. Moreover, the species identified as [M+Na] are ions that appear in the MS during the measurement, and, therefore they should not appear in the column "Compound name".
Table 2: This table is mixed with the neighboring text. Please, see the lines 178-185. This part should be corrected.
Figure 2: This figure is fully illegible, and should be either re-drawn or deleted.
Experimental part: HR-MS and MS/MS experiments are conducted correctly. However, only these experiments cannot lead to the conclusions presented, because additional methods that could prove the structures (eg., NMR) are missing.
Result, Discussion and Conclusion (in general): Based only on the HR-MS and MS/MS data, the presentation of the achieved results is correct. However, the conclusions are not supported with sufficient data and may be misleading. This way of presentation of the results and conclusions brings a question for practical applicability of the method. For example, the authors present oleanolic and ursolic acid. These two triterpenoids have identical number of atoms, and identical molecular mass. Nevertheless, they differ in their structures due to different location of the substituents. Based only on their HR-MS and MS/MS spectra, the structures of those two compounds cannot be determined correctly. To distinguish between the structures of oleanolic and ursolic acid, data from another analytical method, NMR, must be delivered. There are more examples in that table, even if this one is the most visible one.
Evaluation of the manuscript: Based on the above presented criticism, I recommend to reject this manuscript.
Reviewer 2 Report
Comments and Suggestions for Authors
The text below contains comments on manuscript entitled “Development of Tandem Mass Spectral Library for the Detection of Triterpenoids in Plant Metabolome based on Reference Standards”.
To my opinion, the authors should better argument the aim of their study in terms of contribution to the field and what important gaps their study will fill.
The name of the plants Peganum harmala, Camellia sinensis, Aegle marmelos, Adhatoda vasica, and Papaver somniferum that are object of the study should include also the name or the person who described them, when they appear for first time in the text. For example, Peganum harmala L, Camellia sinensis Kuntze etc.
I think that in section 3.1. Chemicals and reagents, the authors should provide the catalogue number and producer of the triterpenoid standards. If the list is too long, it may be delivered as supplementary.
In section 3.3. please specify the rpms of the shaker. The extract were only concentrated or evaporated until dryness also?
Comments on the Quality of English LanguageMinor editing of English language required.